# Update on the Study of Angiogenesis in Surgical Wounds in Patients with Childhood Obesity

**DOI:** 10.3390/biomedicines13020375

**Published:** 2025-02-05

**Authors:** Cristina Chelmu Voda, Ioana Anca Stefanopol, Gabriela Gurau, Maria Andrada Hîncu, Gabriel Valeriu Popa, Olivia Garofita Mateescu, Liliana Baroiu, Mihaela Cezarina Mehedinti

**Affiliations:** 1School for Doctoral Studies in Biomedical Sciences, “Dunarea de Jos” University, 800008 Galați, Romania; cv220@student.ugal.ro (C.C.V.);; 2Department of Morphology and Functional Sciences, “Dunarea de Jos” University, 800008 Galați, Romania; 3Clinical Surgical Department, Faculty of Medicine and Pharmacy, “Dunarea de Jos” University, 800008 Galați, Romania; 4Department of Pediatric Surgery, Sf. Ioan Clinical Emergency Pediatric Hospital, 800487 Galați, Romania; 5Department of Morphology and Functional Sciences, Faculty of Medicine, University of Medicine and Pharmacy of Craiova, 200638 Craiova, Romania; garo2963@yahoo.com; 6Clinical Medical Department, “Dunarea de Jos” University, 800008 Galați, Romania; 7Department of Infectious Diseases, Sf. Cuvioasa Parascheva Clinical Hospital of Infectious Diseases, 800179 Galați, Romania

**Keywords:** angiogenesis, childhood obesity, wound healing, surgical wounds

## Abstract

**Background:** Angiogenesis, the formation of new blood vessels from pre-existing ones, plays a pivotal role in wound healing, particularly in surgical contexts. **Methods and results:** However, this process can be significantly impaired in patients with childhood obesity, resulting in delayed healing and additional complications. The biological process of wound healing is complex, involving angiogenesis, cell proliferation, inflammation, and tissue remodeling. This review aims to explore recent advancements in research on angiogenesis in surgical wounds in patients with childhood obesity, with a focus on growth factors, inflammation, microcirculation, and innovative therapeutic strategies. **Conclusions:** It highlights therapeutic approaches such as the administration of growth factors and the application of biomaterials to enhance angiogenesis.

## 1. Introduction

Childhood obesity has been steadily increasing, affecting approximately 37 million children under the age of five globally in 2022, compared to 33 million in 2000. This rising trend is also evident in the United States and the United Kingdom. While obesity in adults is well documented for its association with surgical and anesthetic complications due to comorbidities, its impact on pediatric surgery remains underexplored and insufficiently understood [1,2]. Childhood obesity represents a significant global public health concern [1,3,4], adversely affecting not only metabolic and cardiovascular health but also the body’s capacity for tissue repair, particularly in surgical wounds.

Angiogenesis, the formation of new blood vessels from pre-existing ones, is a critical process for tissue regeneration and wound healing, ensuring sufficient oxygen and nutrient delivery to areas impacted by surgery or trauma [1,5,6]. This process is highly active in pediatric patients, facilitating rapid and efficient wound healing. However, in children with obesity, angiogenesis can be severely impaired due to oxidative stress and chronic low-grade inflammation—hallmarks of the condition [3].

Xu et al. (2021) demonstrated that elevated levels of pro-inflammatory cytokines, such as TNF-α and IL-6, impair endothelial cell function, inhibiting their ability to expand and migrate for the formation of new blood vessels [6]. Similarly, Cao et al. (2023) reported that chronic inflammation disrupts the balance between proangiogenic and antiangiogenic factors, resulting in insufficient angiogenesis and delayed wound healing [7].

Certelli et al. (2021) highlighted that oxidative stress associated with childhood obesity adversely affects the formation of new blood vessels [4]. Oxidative stress damages endothelial cells and reduces nitric oxide (NO) levels, a crucial vasodilator that supports vascular function [8]. This impairment hinders the natural angiogenic process, thereby prolonging wound healing. Furthermore, Ektoras et al. (2023) observed the reduced activity of VEGF and other growth factors essential for angiogenesis in obese patients, resulting in diminished blood vessel formation [9].

VEGF, a key growth factor, plays a critical role in angiogenesis by stimulating endothelial cell migration and proliferation, thereby facilitating the development of vascular networks. However, pediatric patients with obesity exhibit reduced VEGF expression, compromising the body’s ability to repair blood vessels damaged by surgery or trauma. In addition to suppressing VEGF expression, oxidative stress disrupts VEGF signaling through its receptors (VEGFR), significantly impairing angiogenic capacity.

In addition, childhood obesity impairs the body’s normal inflammatory response, worsening angiogenesis. In patients with childhood obesity, inflammation becomes chronic, adversely affecting vascular function, but normally plays a protective role in the initial stages of wound healing. In 2021, Xu et al. demonstrated that the continued release of pro-inflammatory cytokines is linked to this persistent inflammation. These cytokines inhibit angiogenesis and prevent the body from efficiently repairing damaged tissues [6].

In contrast, Certelli et al. (2021) studied possible treatment methods to improve angiogenesis in childhood obesity [4]. They proposed that the use of biomaterials and the administration of controlled growth factors such as VEGF and PDGF could be useful methods to accelerate the healing process. In preclinical models of childhood obesity, D’Amico et al. (2022) found that the exogenous administration of VEGF significantly increased new blood vessel formation [10]. Thus, given these issues and advances in the understanding of the mechanisms of angiogenesis in childhood obesity, this article aims to provide a review of the mechanisms by which angiogenesis is impaired and to compare recent research on angiogenesis in patients with childhood obesity. In addition, there will be an emphasis on innovative therapeutic approaches, such as gene therapies, growth factor administration, and the use of biomaterials, which have the potential to improve the clinical outcomes of these patients.

Childhood obesity disrupts the body’s normal inflammatory response, further exacerbating impairments in angiogenesis. While inflammation typically plays a protective role in the initial stages of wound healing, in children with obesity it becomes chronic and adversely impacts vascular function. Xu et al. (2021) demonstrated that the persistent release of pro-inflammatory cytokines contributes to this chronic inflammation, which inhibits angiogenesis and impairs the body’s ability to effectively repair damaged tissues [6]. Conversely, Certelli et al. (2021) investigated potential therapeutic strategies to enhance angiogenesis in childhood obesity [4]. Their study suggested that biomaterials and the controlled administration of growth factors, such as VEGF and PDGF, could accelerate the healing process. Supporting this, D’Amico et al. (2022) found that the exogenous administration of VEGF significantly promoted new blood vessel formation in preclinical models of childhood obesity [10].

Given these challenges and advancements, this review explores the mechanisms by which angiogenesis is impaired in childhood obesity and examines recent research in this area. Furthermore, it highlights innovative therapeutic strategies, including gene therapies, growth factor administration, and biomaterial applications, that hold promise for improving clinical outcomes in affected patients.

## 2. Materials and Methods

This review adheres to the PRISMA 2020 reporting guidelines for systematic reviews (Figure 1) [11] and is registered in the International Prospective Register of Systematic Reviews (PROSPERO) under registration number CRD42024606714.

This review synthesizes the latest findings on angiogenesis in surgical wounds among patients with childhood obesity, based on a comprehensive analysis of the available scientific literature. To ensure the inclusion of relevant and up-to-date data, we conducted searches in leading medical databases, including PubMed, Scopus, and Google Scholar, focusing exclusively on peer-reviewed scientific journals.

Each database was searched independently using subject-specific keywords such as ’angiogenesis’, ’childhood obesity’, ’wound healing’, ’surgical wounds’, and ’growth factors’, to address all pertinent aspects of the topic. To ensure the incorporation of recent advancements and methodologies, we restricted the review to articles published between 2018 and 2024. The included studies encompassed both preclinical research (in vitro and animal models) and human clinical investigations that explore the mechanisms of angiogenesis in the context of childhood obesity and surgical wound healing.

Articles that lacked a clear connection to angiogenesis in the context of childhood obesity and surgical wounds were excluded from the analysis based on the predefined exclusion criteria. Furthermore, to ensure the inclusion of only current and relevant research, we excluded articles based on unverified hypotheses or those published prior to 2018. The selected studies met our inclusion criteria, adhering to strict methodological standards and focusing on data collected within a preclinical context.

An additional filter was applied during the selection process to ensure that only studies utilizing rigorous research methodologies, such as randomized controlled trials, meta-analyses, and systematic reviews, were included. All references were imported into Zotero, and duplicates were removed.

The search results are illustrated in a flow diagram (Figure 1), which displays the initial 15,648 results from three sources, reduced to 2861 articles after deduplication for title and abstract screening. Following the application of the inclusion and exclusion criteria, 2688 articles were deemed irrelevant, leaving 173 articles for full-text screening. After reviewing the full-text articles, 114 were excluded, resulting in 59 studies being retained for the final analysis.

## 3. Mechanisms of Angiogenesis and Its Role in Wound Healing

### 3.1. The Role of Growth Factors in Angiogenesis

The release of pro-angiogenic factors, such as Platelet-Derived Growth Factor (PDGF) and Vascular Endothelial Growth Factor (VEGF), is triggered by tissue hypoxia, which commonly occurs in surgical wounds [12]. Both of these factors are critical for the initiation and maintenance of angiogenesis. As noted by Wang et al. (2022), VEGF plays a pivotal role in promoting endothelial cell migration and proliferation, thereby facilitating the formation of new blood vessels [13]. Recent research indicates that hypoxia-induced autophagy not only stimulates angiogenesis but also provides protective effects for endothelial cells [14,15]. Additionally, D’Amico et al. (2022) underscored the importance of PDGF in stabilizing newly formed blood vessels through the recruitment of pericytes [10].

One of the key growth factors involved in the angiogenesis process is Vascular Endothelial Growth Factor (VEGF). Xu et al. (2021) demonstrated that, under hypoxic conditions, the expression of VEGF is regulated by Hypoxia-Inducible Factor 2α (HIF-2α). In addition to HIF-2α, other transcription factors, such as HIF-1α, also influence VEGF expression in response to hypoxia, thereby enhancing the angiogenic response [6].

Certelli et al. (2021) observed that VEGF activity is heightened in healthy individuals, facilitating faster blood vessel regeneration and efficient wound healing [4]. In contrast, persistent inflammation, oxidative stress, and the inhibition of VEGF expression and activity may occur in patients with childhood obesity, leading to impaired angiogenesis and delayed wound healing [4]. The chronic inflammation associated with obesity adversely impacts VEGF signaling, diminishing the body’s capacity to repair damaged tissues.

Angiogenesis in childhood obesity may also be influenced by the disruption of other growth factors and cytokines. Research has shown that VEGF signaling and endothelial function can be impaired by elevated levels of Tumor Necrosis Factor-alpha (TNF-α) and Interleukin-6 (IL-6) [16]. Additionally, adipokines such as leptin and adiponectin, which are secreted by adipose tissue, play complex roles in the angiogenesis process [17,18]. Fluctuations in the levels of these adipokines can lead to angiogenic dysfunctions [16].

Leptin, in particular, has been shown to enhance the secretion of inflammatory cytokines, including TNF-α, IL-6, and IL-12. Furthermore, inflammatory stimuli such as TNF-α and IL-1 promote leptin expression in both the circulation and adipose tissue, establishing a feedback loop that amplifies inflammation. This feedback mechanism underscores how leptin contributes to low-grade inflammation, as pro-inflammatory mediators increase leptin expression, alongside other acute-phase reactants that perpetuate chronic inflammation [14,19,20,21].

### 3.2. The Role of HIF-2α in Angiogenesis

Hypoxia-Inducible Factor-2α (HIF-2α) is a transcription factor that plays a pivotal role in regulating angiogenesis under hypoxic conditions by stimulating the expression of VEGF and other proangiogenic genes. In a study by Xu et al. (2021), HIF-2α was shown to not only induce angiogenesis but also contribute to the maturation and stabilization of newly formed blood vessels, both of which are essential for effective wound healing [6]. Experimental models further demonstrated that HIF-2α-deficient mice exhibited impaired vascular regeneration and slower wound healing [19].

Recent clinical research has indicated that patients with childhood obesity exhibit reduced levels of HIF-2α in affected tissues, which compromises their ability to heal wounds (Certelli et al., 2021) [4]. The oxidative stress and chronic inflammation commonly associated with obesity can negatively impact HIF-2α, leading to a reduction in the transcriptional activity and expression of VEGF [18,22].

Shaabani et al. (2022) demonstrated that high oxidative stress can induce HIF-2α degradation, thereby limiting the angiogenic response to hypoxia [23]. In light of these findings, a promising therapeutic strategy could involve the use of antioxidants or agents that stabilize HIF-2α to enhance angiogenesis and wound healing [13,24,25]. Furthermore, Zou et al. (2024) suggested that interventions aimed at reducing systemic inflammation could restore normal HIF-2α function, potentially improving angiogenesis and wound healing in patients with childhood obesity [26].

### 3.3. Tenascin-C as a Key Regulator of Angiogenesis

Tenascin-C (TNC) is a matrix glycoprotein that plays a crucial role in angiogenesis and tissue regeneration. Wang et al. (2022) emphasized TNC’s involvement in the activation of proangiogenic pathways, particularly through its interactions with Vascular Endothelial Growth Factor (VEGF) and αvβ3 integrins [13]. TNC is essential during the early stages of wound healing, where it influences cell adhesion, migration, and the remodeling of the extracellular matrix.

In pediatric patients, the appropriate expression of TNC is crucial for the formation of a stable vascular network and effective tissue regeneration. However, the dysregulation of TNC expression can result in impaired angiogenesis and delayed wound healing in patients with childhood obesity, as highlighted by Certelli et al. (2021) [4]. Chronic inflammation, a hallmark of obesity, can alter the structure and function of TNC, affecting its interactions with endothelial cells and growth factors. Wang et al. (2022) suggested that TNC may be overexpressed in conditions of chronic inflammation [13], leading to extracellular matrix disorganization and the inhibition of conventional angiogenesis.

Furthermore, D’Amico et al. (2022) found that TNC can influence macrophage polarization toward the anti-inflammatory M2 phenotype, which promotes tissue regeneration [10]. However, in childhood obesity, the predominant presence of pro-inflammatory M1 macrophages limits the beneficial effects of TNC on angiogenesis, thus exacerbating angiogenic dysfunction and impairing wound healing [13,27].

### 3.4. Childhood Obesity and Angiogenesis Dysfunction

Through endothelial activation (EA) and chronic inflammation, childhood obesity directly impairs angiogenesis and wound healing. Endothelial activation reduces blood circulation and oxygen delivery to affected tissues, disrupting the delicate balance between vasodilators and vasoconstrictors. As a result, the body is unable to efficiently transport the nutrients and oxygen required for tissue repair [28].

Pro-inflammatory factors, along with the increased secretion of molecules such as leptin, TNF-α, and VEGF, negatively impact angiogenesis in individuals with obesity. Although these molecules are typically involved in promoting balanced blood vessel formation, in obese children, the hyperactivity of VEGF and other angiogenic factors can lead to abnormal blood vessel development. This dysfunction results in disorganized vascularization and inadequate wound healing, thereby increasing the risk of infections and poor postoperative recovery.

In response to tissue damage, hypoxia develops at the affected site due to vascular structural destruction. This drop in oxygen levels activates the transcription factor HIF-1 (Hypoxia-Inducible Factor), which initiates cellular mechanisms to detect and adapt to the hypoxic conditions. The HIF-1 transcription factor plays a central role in regulating the cellular response to hypoxia. Under low-oxygen conditions, the active HIF-1α subunit is triggered and induces the production of VEGF-A (Vascular Endothelial Growth Factor-A), a key proangiogenic mediator that governs the vascular response to hypoxia. VEGF-A facilitates the rapid formation of a dense, new vascular network, which is critical for tissue repair [29].

In adipose tissue, hypoxia-inducible factor (HIF-1α) expression is stimulated when inadequate oxygen supply fails to meet the tissue’s increased demand. This results in compensatory angiogenesis, though of a low quality. Such angiogenesis exacerbates wound conditions by creating fragile, poorly functioning blood vessels, thereby hindering proper healing [30].

Increased oxidative stress and persistent low-grade inflammation, both directly linked to childhood obesity, significantly impact angiogenesis. Shaabani et al. (2022) demonstrated that pro-inflammatory cytokines such as IL-6 and TNF-α reduce the production of nitric oxide (NO), a critical mediator for vasodilation and endothelial function [23]. Reduced NO levels lead to endothelial dysfunction and impair the formation of new blood vessels, as NO plays a vital role in angiogenesis [23].

Hypertrophied adipocytes in obese adipose tissue produce reactive oxygen species (ROS) and additional inflammatory cytokines, contributing to a sustained inflammatory response. While inflammation is an essential step in the healing process, chronic inflammation in obesity disrupts angiogenesis by damaging endothelial cells. The balance between pro- and anti-angiogenic molecules, such as VEGF and endostatin, normally regulates angiogenesis under healthy conditions. However, in obesity, this balance is disrupted, leading to a compromised vascular environment that impairs effective tissue repair.

The accumulation of reactive oxygen species (ROS), which leads to oxidative stress, has a detrimental effect on endothelial cells [25,31]. This damage occurs through endothelial cell apoptosis and the inhibition of their migration and growth (Xu et al., 2021) [6]. Insulin resistance, along with increased levels of endothelin-1, a potent vasoconstrictor that impairs blood flow and angiogenesis, further contributes to endothelial dysfunction in childhood obesity. These metabolic and hormonal changes disrupt the angiogenic balance, prolonging wound healing times and increasing the likelihood of complications [6,32,33,34].

Additionally, the adipokine balance is altered in childhood obesity. Obradovic et al. (2021) demonstrated that elevated leptin levels can have a dual effect on angiogenesis. However, in the presence of chronic inflammation, the negative impact predominates, resulting in vascular dysfunction. On the other hand, adiponectin, an adipokine with anti-inflammatory and proangiogenic properties, is typically reduced in obesity, exacerbating angiogenic problems.

Key proangiogenic factors, such as VEGFA and fibroblast growth factor 2 (FGF2), play crucial roles in promoting capillary formation and stimulating endothelial cell migration, essential processes for the formation of new blood vessels required for wound regeneration. In children with obesity, however, angiogenic activity is often insufficient and dysregulated, limiting vascular formation. This can result in tissue ischemia, prolonging the healing process. Elevated levels of inflammation and pro-inflammatory cytokines in adipose tissue contribute to this impaired angiogenic response, reducing the ability of wounds to heal rapidly. Furthermore, inadequate blood vessel proliferation can hinder wound revascularization, negatively affecting tissue regeneration quality and increasing the risk of infection.

Thus, persistent inflammation, oxidative stress, dysregulation of growth factors, and alterations in the function of matrix proteins such as Tenascin-C contribute to the complex causes of angiogenesis dysfunction in patients with childhood obesity. To develop therapeutic strategies that enhance wound healing and reduce complications in these patients, it is crucial to gain a comprehensive understanding of these mechanisms.

Exploring the processes and key factors involved in wound healing is essential for identifying new therapeutic approaches aimed at improving angiogenesis and granulation tissue formation. These factors are particularly important for children with varying body weights (obese compared to normal weight), as well as for wounds located on different areas of the body, including the upper limbs, lower limbs, and abdomen. According to Ciprandi et al., pediatric skin is characterized by incomplete microvascularization and a weak connection between the epidermis and dermis [1]. This makes children’s skin more vulnerable to injury and infection due to its fragility. Local hypoxia, often present in obese children due to the adipose layer, further impedes angiogenesis and delays wound closure. Gao et al. (2021) have highlighted that obese pediatric patients are at a higher risk of complications and delayed healing, underscoring the importance of addressing these factors in treatment strategies [5].

The efficiency of angiogenesis plays a critical role in accelerating tissue repair and reducing the risk of complications, especially in cases of persistent wounds influenced by predisposing factors such as childhood obesity. Granulation tissue, composed of a network of new capillaries, fibroblasts, and structural proteins, is vital for wound closure and the prevention of infection. Angiogenic activity, growth factors like VEGF (vascular endothelial growth factor), and microRNAs all contribute to the formation and function of granulation tissue.

Obese children experience slower angiogenesis and are at a higher risk for prolonged inflammation compared to children of normal weight [29,35,36]. These factors may hinder the proper formation of granulation tissue. Studies have shown that obesity impairs the healing process of surgical wounds in children, particularly by influencing the inflammation and proliferation phases, both of which are essential for the formation and angiogenesis of granulation tissue. Obese children typically exhibit a more intense and prolonged inflammatory response, which delays the transition to the proliferative phase, consequently hindering healing. In contrast, children of normal weight typically coordinate these stages more effectively, allowing them to heal faster and form higher-quality granulation tissue (Figure 2).

The figure provides a visual representation of the healing aspects in normal-weight versus obese pediatric patients, comparing several key parameters of wound healing: Healing Rate (days): The time taken for complete wound closure is generally longer in obese children compared to normal-weight children, reflecting delayed healing processes. Granulation Tissue Formation (%): Obese children show reduced formation of granulation tissue, which is crucial for wound closure, as compared to normal-weight children. Infection Risk (%): The likelihood of infection during the healing process is higher in obese children, attributed to the prolonged inflammation and insufficient angiogenesis. Common Complications (%): Obese children experience a higher percentage of complications, such as delayed healing or wound dehiscence. Vascularization Efficiency (%): The formation of new blood vessels in the wound area is less efficient in obese children, limiting oxygen and nutrient supply to the tissue. Tissue Oxygenation (%): Oxygen levels at the wound site are often lower in obese children, which hinders tissue repair and regeneration. Inflammatory Response Level (%): Obese children exhibit a higher inflammatory response, which delays the transition to the proliferative phase of healing. Pain and Mobility Recovery (days): Recovery from pain and mobility is slower in obese children due to the prolonged inflammatory phase. Typical Healing Duration (weeks): The overall duration of healing is longer in obese children due to the combined effects of inflammation, reduced angiogenesis, and slower granulation tissue formation.

The blue bars represent normal-weight pediatric patients, while the orange bars represent obese pediatric patients. The graph highlights the inefficiencies in healing and the increased risk of complications in obese children, underscoring the impact of obesity on wound healing and recovery.

In normal-weight children, granulation tissue in the upper limbs forms rapidly, supported by efficient blood circulation, allowing for healing within 7–10 days. In contrast, in obese children, the accumulation of fatty tissue leads to edema (fluid retention) and chronic inflammation, which slows down the healing process. These conditions increase the likelihood of infection and the formation of hypertrophic scars (raised, thickened scars). Studies indicate that obese children face a greater risk of delayed healing due to these factors [5,31,37].

Regarding the lower limbs, the natural peripheral circulation is slower due to mechanical pressure (often caused by excess weight) and persistent inflammation. Lower limb injuries, including those resulting from outdoor accidents, are especially problematic for obese children. Gao et al. note that these children are at a significantly higher risk of poor healing outcomes in such cases.

Additionally, wound size plays a key role in healing time. Wounds greater than 3 cm in length, regardless of the child’s body weight, are associated with longer healing times. Moreover, age is another important factor; the risk of poor healing increases by 34 percent with each additional year of age [5]. Therefore, both obesity and age contribute significantly to the challenges in wound healing in pediatric patients.

In obese children, the wound healing process is significantly delayed, which increases the risk of chronic ulceration and infection. This makes it more difficult for wounds to heal compared to normal-weight children, in whom granulation tissue forms more rapidly, typically within 7–14 days, depending on the severity of the wound. Normal-weight children have a reduced fat layer in areas like the abdomen, which facilitates faster healing and a lower risk of infection. However, in obese children, the fat layer in the abdominal region hampers blood circulation, which slows the healing process and increases the risk of postoperative infections and wound dehiscence (wound reopening), especially after surgery.

Several studies highlight the impact of obesity on wound dehiscence rates, with obese pediatric patients being twice as likely to experience wound dehiscence compared to their normal-weight counterparts [2,12,38,39,40]. Additionally, prolonged surgical duration is a well-established factor that increases the risk of surgical complications. Longer surgeries lead to longer exposure to anesthesia, higher blood loss, more tissue trauma, and greater inflammation. These factors, in combination with fluid and electrolyte imbalances, contribute to a higher incidence of complications like infections, erythema (skin redness), necrosis (tissue death), seromas (fluid accumulation), hematomas (blood collection), and delayed wound healing.

These challenges highlight the importance of addressing obesity and its associated risks in pediatric surgical patients to improve healing outcomes and reduce complications.

As the comparative graph (Figure 3) shows, wound healing in pediatric patients shows significant differences depending on body weight and the location, size, and depth of the lesions. Children with a normal weight have more efficient healing processes due to better vascularization, a smaller fat layer, and a better regulated inflammatory response. In contrast, obese children greatly delay the healing process due to factors such as chronic inflammation, mechanical pressure, and deficient angiogenesis.

Small, superficial wounds tend to heal more rapidly in normal-weight children, with healing efficiency exceeding 80%. In contrast, this efficiency significantly decreases in obese children, especially for larger, deeper wounds, where healing efficiency drops to below 50%.

Location also plays an important role, with abdominal and lower limb wounds having delayed healing in obese children due to the extensive fat layer and poor peripheral circulation.

The wound healing process is negatively affected by obesity both through local mechanisms, such as angiogenesis deficiency and prolonged inflammation, and through systemic effects, such as metabolic imbalances and increased mechanical pressure on tissues. Normal-weight children, on the other hand, benefit from a faster healing process, characterized by rapid transitions between the phases of inflammation, proliferation, and remodeling. The need for individualized approaches to wound management in obese children is emphasized by these differences. These approaches should include decreasing persistent inflammation, stimulating angiogenesis, and using modern dressing technologies to reduce the risk of complications [1,5,35].

### 3.5. Therapeutic Interventions to Improve Angiogenesis

Local or systemic administration of VEGF and PDGF has been shown to improve angiogenesis and accelerate wound healing in patients with childhood obesity. D’Amico et al. (2022) demonstrated that exogenous administration of VEGF has the potential to stimulate the creation of vascular networks in skin affected by persistent wounds [10]. In addition, Wang et al. (2022) found that using PDGF together with biomaterials can help the newly formed vessels to be more stable, thus preventing their regression [13].

In hypertrophied adipose tissue, commonly observed in obese children, hypoxia serves as a significant aggravating factor in the wound healing process. Hypoxic conditions activate the transcription factor HIF-1α, which in turn stimulates the production of VEGF and other angiogenic and inflammatory mediators.

Chronic hypoxia stimulates inflammation, and this continuous inflammatory environment prevents the proper formation of the granulation tissue necessary for wound closure. Thus, instead of hypoxia effectively stimulating angiogenesis, it contributes to insufficient vascular regeneration and can cause cell death and poor-quality scar tissue formation, increasing the risk of complications in the healing process.

Biomaterials designed for the controlled delivery of growth factors, such as fibrin hydrogels, represent an innovative approach in wound healing. These hydrogels facilitate the constant and controlled release of growth factors, ensuring their sustained presence at the wound site. Certelli et al. (2021) [4] demonstrated that fibrin hydrogels stimulate angiogenesis and help prevent postoperative complications by providing a stable environment for growth factor release. Furthermore, these biomaterials protect growth factors from rapid degradation, thereby enabling controlled and effective angiogenesis, which is essential for optimal wound healing [9,18,27].

MicroRNAs are, in addition to the involvement of growth factors, a new area of research in the regulation of angiogenesis. They can both stimulate and inhibit angiogenesis, offering new therapeutic approaches [41]. The study proposed by Tejedor et al. (2024) [37] demonstrate that, compared to the specific factors used, a combination of mRNA for the growth factors VEGF-A and FGF1, encapsulated in lipid nanoparticles, significantly improves angiogenesis and wound healing in diabetic mice. The stimulation of skin regeneration by FGF1 and the formation of blood vessels assisted by VEGF-A lead to the activation of vital healing processes [37].

Nanotechnology uses nanomaterials such as hydrogels and nanofibers to aid in wound healing through hemostasis, antibacterial action, and inflammation control. Metal nanoparticles, such as gold, silver, and zinc have antibacterial properties, and polymer nanofibers that mimic the extracellular matrix (ECM), aid in cell adhesion and proliferation. However, to prevent the eventual toxicity of metal nanotherapeutics, caution and the optimization of the structure of nanomaterials are necessary [42].

Another promising treatment method for improving angiogenesis in patients with childhood obesity is gene therapy. In preclinical models, Xu et al. (2021) demonstrated that altering the expression of genes involved in angiogenesis, such as VEGF and PDGF, could accelerate wound healing [6]. However, due to safety and gene expression control issues, the clinical use of this therapy is still limited.

Revascularization is necessary in wound treatment to reduce scarring and improve patients’ quality of life [40]. Wounds in obese children are difficult to heal due to the persistent inflammatory state and metabolic imbalances, which hinder angiogenic and cell regeneration processes [36]. Recent research shows that obese children have more chronic wounds and delayed healing due to decreased angiogenesis. In the inflammatory environments caused by obesity, VEGF and FGF do not function well in this population, and vascular regeneration at the site of injury is decreased.

To enhance angiogenesis in children with obesity, innovative biomaterials, including injectable hydrogels and compounds such as carboxymethyl chitosan, have been developed. These materials not only maintain an optimal moist wound environment but also enable the controlled release of growth factors, thereby promoting effective healing. Additionally, cell therapies utilizing exosomes and mesenchymal stem cells (MSCs) have shown promise in supporting vascular proliferation and controlling inflammation. These therapies aim to foster more efficient healing by addressing the specific challenges faced by obese children, particularly in relation to angiogenesis and wound recovery [41,42].

## 4. Discussion

A number of factors influence angiogenesis, which is an essential process in the healing of surgical wounds, such as the general health of the patient, the activity of pro-angiogenic factors, and the existence of diseases such as childhood obesity. Research has focused in recent years on identifying molecular mechanisms and treatment modalities that can improve angiogenesis, particularly in obese patients and children.

The most important growth factors associated with angiogenesis, VEGF and PDGF, play a direct role in the formation and stabilization of new blood vessels [1,4]. Endothelial cell proliferation and migration are driven by VEGF, which are essential for the formation of vascular networks in tissues that have been damaged by surgery or trauma. In a recent study, Certelli et al. (2021) showed that the presence of oxidative stress and systemic inflammation have a direct impact on VEGF activity [4]. VEGF helps support angiogenesis in pediatric patients, which allows for faster and more efficient wound healing.

In contrast, persistent inflammation and endothelial cell dysfunction reduce VEGF activity in patients with childhood obesity. High levels of pro-inflammatory cytokines, such as TNF-α and IL-6, impede endothelial function and decrease the bioavailability of nitric oxide (NO), which is essential for vasodilation and promoting angiogenesis. The ability of endothelial cells to respond to pro-angiogenic signals is impaired by reduced NO, which greatly delays surgical wound healing.

PDGF, another important growth factor involved in angiogenesis, helps stabilize new blood vessels by attracting pericytes and strengthening vascular structures created by VEGF [4]. D’Amico et al. (2022) demonstrated that PDGF signaling is disrupted in childhood obesity [10]. This causes the vascular networks to become more unstable and more susceptible to regression. Thus, obese patients delay wound healing [30] and are more susceptible to postoperative complications such as infections and chronic wound development due to VEGF and PDGF dysfunction.

Some genes that are involved in the development of new blood vessels, including VEGF, are triggered by HIF-2α (Hypoxia-Inducible Factor-2α). This is necessary for the hypoxic response. In tissue hypoxia, VEFG expression is elevated by HIF-2α. Angiogenesis in surgical wounds is aided by this. HIF-2α is more selective than HIF-1α in the last phases of angiogenesis and is necessary for the maturation and stabilization of newly formed blood vessels, according to research conducted in 2021 by Xu et al. [6,16].

During wound healing in children, HIF-2α is expressed at high levels, which helps in rapid vascular regeneration. Conversely, chronic inflammation and oxidative stress impair HIF-2α activity in childhood obesity patients, which reduces angiogenesis and delays wound healing. Certelli et al. (2021) found that in childhood obesity, the inhibition of HIF-2α activity is linked to decreased VEGF expression [4]. This explains the difficulties with vascular regeneration in these children [43].

These findings indicate that therapies aimed at increasing HIF-2α expression could be a solution for patients with childhood obesity. This could lead to improved blood vessel regeneration and surgical wound healing [15]. However, more clinical trials are needed to evaluate the efficacy and safety of potential HIF-2α-modifying treatments in this setting [6].

Tenascin-C (TNC) is a matrix glycoprotein that participates in inflammation, regeneration, and angiogenesis processes. Recent studies have shown that TNC performs two different functions [13]. First, it helps to polarize macrophages towards an anti-inflammatory M2 phenotype, and second, it stimulates angiogenesis by interacting with VEGF and αv3 integrins. TNC is vital for stimulating angiogenesis in the initial stages of wound healing, particularly by creating a proangiogenic and anti-inflammatory environment [13].

High levels of TNC help heal more effectively and regenerate vascular tissue faster. Conversely, altered cell signaling, persistent inflammation, and childhood obesity impair TNC expression.

### 4.1. Challenges of Angiogenesis in Pediatric Patients with Childhood Obesity

The capacity for vascular regeneration is severely impaired by the chronic low-grade inflammation that is a feature of childhood obesity [43]. Pro-inflammatory cytokines, such as TNF-α and IL-6, directly influence endothelial cells by inhibiting VEGF-mediated proangiogenic signaling. This reduces the capacity for vascular regeneration. A continued vasoconstriction and a decrease in blood flow to the affected areas is caused by the reduction in NO. This prevents the body from receiving nutrients and oxygen from surgical wounds [44].

In addition, persistent inflammation prevents macrophages from functioning, which are vital to the healing process. Macrophages switch from the pro-inflammatory M1 phenotype to the anti-inflammatory M2 phenotype under normal conditions, which aids in angiogenesis and tissue regeneration [27]. Patients with childhood obesity experience this transition, which maintains an ongoing inflammatory state and impedes the healing process. Certelli et al. (2021) [4] found that, in these patients, chronic inflammation prevents macrophage polarization towards the M2 phenotype. This reduces angiogenesis and reduces wound healing capacity [4].

Another important factor that has a negative impact on angiogenesis in patients with childhood obesity is oxidative stress, which is characterized by excessive production of reactive oxygen species (ROS). In 2021, Xu et al. [6] found that ROS directly affect endothelial cells, preventing them from migrating and growing, preventing the development of new blood vessels. In addition, oxidative stress destroys endothelial cell proteins and lipids, causing the cells to function more poorly and ultimately inhibiting angiogenesis [6,13].

Similarly, D’Amico et al. (2022) demonstrated that ROS inhibit VEGF-mediated proangiogenic signaling [10]. This reduces VEGF activity and prevents the body from initiating and maintaining angiogenesis in surgical wounds. The risk of developing non-functional scar tissue and delaying wound healing increases when oxidative stress prevents vascular regeneration. Compared to healthy children, pediatric patients with childhood obesity produce ROS in a much higher amount than healthy children. This explains why these patients are more susceptible to postoperative complications and the development of chronic wounds [13].

Another significant issue in dysfunctional angiogenesis in childhood obesity patients is defective growth factor signaling. Certelli et al. (2021) [4] found that VEGF and PDGF expression is significantly reduced in these patients. This means that they cannot build and stabilize new blood vessels [4]. Pro-inflammatory cytokines inhibit the normal function of endothelial cells and reduce their sensitivity to growth signals [10,45].

Furthermore, Corvera et al. (2022) discovered that the expression of VEGF and PDGF receptors is changed in obese patients with chronic inflammation [45]. This stops proangiogenic signaling pathways from opening properly. Because of this, VEGF and PDGF signaling is ineffective even in the presence of sufficient quantities, which causes incomplete and unstable vascular networks to emerge.

### 4.2. Therapeutic Interventions to Improve Angiogenesis

Vascular endothelial growth factor (VEGF) and platelet-derived growth factor (PDGF) are essential for promoting angiogenesis and wound healing. VEGF plays a crucial role in the development of new blood vessels by stimulating the migration and proliferation of endothelial cells. Certelli et al. (2021) [4] demonstrated that VEGF interacts with receptors on the surface of endothelial cells, triggering intracellular signaling pathways that activate the angiogenic process. This process is vital for the regeneration of tissue and the efficient healing of wounds, particularly in cases where angiogenesis is impaired, such as in obese patients [4].

VEGF levels are naturally high in children, which helps them heal faster from injuries than adults. However, chronic low-grade inflammation and oxidative stress significantly reduce VEGF activity in patients with childhood obesity. This affects the body’s ability to build new blood vessels and repair damaged tissue. Pro-inflammatory cytokines, such as TNF-α and IL-6, negatively influence VEGF function. This happens by inhibiting VEGFR-2 receptors, which are responsible for generating proangiogenic signals.

PDGF, another essential factor in the wound healing process, supports angiogenesis by stabilizing newly formed blood vessels. PDGF seeks out pericytes, which are cells essential for the stabilization and maturation of capillaries, after VEGF begins blood vessel formation. Blood vessels are fragile and susceptible to regression in the absence of PDGF [46]. D’Amico et al. (2022) found that patients with childhood obesity have decreased PDGF signaling [10]. This makes the new vascular networks unstable and delays healing.

Growth factor therapy, especially the combination of VEGF and PDGF, has been tested to improve angiogenesis in patients with chronic wounds in preclinical and clinical studies. Wang et al. (2022) found that the establishment of stable and functional vascular networks was accelerated by the simultaneous administration of VEGF and PDGF in patients with chronic wounds [13]. This therapy improved tissue regeneration and reduced the risk of postoperative complications, especially in obese pediatric patients, where inflammation and oxidative stress are essential [13,46,47].

However, controlling the doses and duration of administration is one of the biggest problems with growth factor therapy. Certelli et al. (2021) showed that high VEGF can lead to unstable or non-functional blood vessels, which increases the risk of aberrant angiogenesis [4]. Therefore, to prevent these side effects and ensure efficient and stable angiogenesis, the doses of VEGF and PDGF must be carefully adjusted [4,27,48,49,50].

Biomaterials that allow for the controlled and sustained release of growth factors have been created to improve growth factor therapy and reduce the risk of complications. Fibrin hydrogels are an innovative biomaterial that has proven highly effective in applying growth factors to the wound site in a controlled manner. According to Wang et al. (2022), fibrin hydrogels have the ability to serve as a reservoir for VEGF and PDGF, gradually releasing these growth factors according to the physiological requirements of the targeted area [13].

The ability of hydrogels to protect growth factors from rapid degradation in the biological environment is a great advantage. Certelli et al. (2021) showed that angiogenesis becomes more efficient when VEGF and PDGF are gradually released through hydrogels [4]. This makes the formation of blood vessels more stable and efficient [20,40]. This is because hydrogels maintain a constant level of growth factors at the wound site, preventing fluctuations that could lead to aberrant or insufficient angiogenesis.

Biomaterials such as fibrin hydrogels can also be tailored to the individual needs of each patient. D’Amico et al. (2022) noted the possibility of modifying these biomaterials to release growth factors over a longer period of time [10,19]. This is important for patients with persistent injuries or conditions such as childhood obesity [21], which impairs tissue regeneration and slows the growth process.

Hydrogels optimize the delivery of growth factors and can be used in conjunction with other bioactive molecules that promote healing, such as anti-inflammatory peptides or antioxidants. Wang et al. (2022) demonstrated that introducing anti-inflammatory molecules into hydrogels can help decrease persistent inflammation in children with childhood obesity [13]. This facilitates angiogenesis and tissue regeneration.

However, the use of biomaterials also involves problems. To do this, the biomaterial can be incorporated into the host tissue without causing an adverse immune reaction. Certelli et al. (2021) showed that although fibrin hydrogels are highly biocompatible, rejection or the occurrence of an unwanted inflammatory reaction are still possible [4]. Therefore, before the widespread clinical use of biomaterials, it is essential that they are rigorously tested.

The application of biomaterials for the controlled delivery of growth factors is a promising approach for the treatment of surgical wounds in children with childhood obesity and adolescents. It has been demonstrated that fibrin hydrogels are incredibly successful at promoting angiogenesis and enhancing therapeutic results. To optimize these biomaterials’ long-term safety and efficacy, more research is required.

A new direction in the treatment of surgical wounds, gene therapy, has the potential to revolutionize the way angiogenic dysfunctions are treated in patients with childhood obesity. The aim of these treatments is to alter the expression of genes involved in angiogenesis, such as VEGF and PDFG. This is achieved by using viral vectors or gene editing methods such as CRISPR/Cas9 [13,23].

Studies have shown that the gene transfer of VEGF via viral vectors can stimulate angiogenesis in hypoxic areas and accelerate wound healing. In preclinical models of childhood obesity, Xu et al. (2021) demonstrated that gene therapy improved blood vessel stability and angiogenesis [6]. In addition, D’Amico et al. (2022) demonstrated that gene therapy can improve proangiogenic signaling in obese patients, where inflammation and oxidative stress impede the natural capacity for vascular regeneration [10].

However, there are a number of significant problems that arise when gene therapy is implemented in practice. The risk of immune reactions against viral vectors used to deliver therapeutic genes is one of them.

Altered gene expression is another significant challenge. Certelli et al. (2021) point out that precise regulation of the expression of genes involved in angiogenesis is essential in gene therapy [4]. For example, excessive VEGF proliferation can lead to strange, unstable, or non-functional blood vessels. In addition, there is a greater likelihood that the altered genes will be expressed in an uncontrolled manner, which increases the likelihood of complications such as the development of vascular tumors.

The use of non-viral vectors or precise gene editing techniques such as CRISPR/Cas9 can solve these problems without endangering viral vectors [23]. According to Wang et al. (2022), one can use CRISPR to control the expression of proangiogenic genes in a more precise and controlled way, thereby reducing the risk of complications and improving the efficiency of treatment [13,23].

Gene therapy is a promising approach to treat angiogenesis dysfunction in children with childhood obesity, despite obstacles. Future research should focus on optimizing the control of altered gene expression and creating safer and more efficient ways to deliver therapeutic genes.

With the advantages of low immunogenicity, high stability, biodegradability, and barrier crossing ability, mesenchymal stem cell-derived extracellular vesicles (EVs) have significant potential to aid in wound healing. EVs can be used to regenerate blood vessels, nerves and hair follicles and help early and scarless healing by accelerating hemostasis, improving inflammation, stimulating endothelial cell and fibroblast proliferation, preventing extracellular cell (ECM) overproduction, improving tissue remodeling, and preventing scar formation [12,32,33].

### 4.3. The Use of Multifunctional Biomaterials in Wound Healing

Biomaterials, particularly injectable hydrogels, have emerged as a promising therapy option for surgical wounds because of their capacity to deliver additional bioactive molecules and regulated growth factors straight to the wound site. Researchers have studied the use of multifunctional hydrogels for healing diabetic wounds and found that they better stimulate angiogenesis and reduce inflammation [9,51]. Hydrogels serve as reservoirs for the slow and sustained release of VEGF and PDGF. This ensures the stability of the blood vessels formed and prevents their regression [7,9,18,50].

Multifunctional hydrogels can also simultaneously release different bioactive molecules such as antioxidants and antimicrobials. These bioactive molecules are essential for the persistent inflammation that occurs in obese children. In 2023, Cao et al. revealed that self-healing hydrogels had the ability to protect tissues against infection and induce angiogenesis by the intelligent release of VEGF in the presence of inflammatory stimuli [4,6,7,52].

Cao et al. (2023) demonstrated that injectable hydrogels containing micelles can treat persistent wounds in patients with diabetes and obesity by providing a controlled release of VEGF and antimicrobial drugs [7]. This technology improved the formation of stable vascular networks and reduced healing time.

A promising way to stimulate angiogenesis is by stimulating the expression of the VEGF and PDGF genes, which are essential for blood vessel formation. According to Xie et al. (2022), the modulation of microRNAs has the potential to restore the ability of endothelial cells to respond to proangiogenic growth factors [34,50,53]. This provides a solution to the signaling deficiencies seen in childhood obesity.

MicroRNAs control the expression of genes involved in angiogenesis, and their disrupted expression can lead to angiogenic dysfunctions [23,50]. Angiogenesis in chronic wounds, including obesity, can be significantly enhanced with therapeutic microRNAs, according to Shaabani et al. (2022) [23].

The application of advanced biomaterials for the delivery of growth factors presents numerous challenges. A major difficulty lies in optimizing the dosage and release rate, as excessive dosage can lead to the formation of non-functional blood vessels and severe inflammatory reactions, while insufficient dosage fails to support adequate angiogenesis. Furthermore, even biomaterials considered biocompatible, such as collagen, keratin, fibrin-based hydrogels, or chitosan, can trigger unwanted immunological reactions or sensitization. The integration of these materials must be carefully aligned with the phases of wound healing (inflammation, proliferation, remodeling) to ensure optimal spatial and temporal synchronization, thus preventing therapy failure or excessive scarring. In addition, the high costs and complex logistics, including the infrastructure required to produce advanced biomaterials (such as those with nanoparticles or keratin-based structures), limit the widespread adoption of these technologies in many medical settings [42].

The use of viral vectors in gene therapy involves significant immunological risks and challenges in dose regulation. The overexpression of growth factors such as VEGF can lead to dysfunctional blood vessel formation, and maintaining pro-angiogenic gene expression within a therapeutic window is difficult, increasing the risk of abnormal neovascularization [7,18,50].

Growth factors such as FGF, VEGF, and PDGF are susceptible to proteolytic degradation and diffusion away from the wound site, reducing their therapeutic efficacy. Prolonged administration or high doses of these factors can lead to allergic reactions and excessive inflammation. In this context, more efficient delivery systems such as nanoparticle encapsulation or hydrogels are needed. However, these technologies involve complex manufacturing procedures, high costs, and increased safety requirements [4,42].

Mesenchymal stem cells (MSCs) are multipotent stem cells derived from the mesoderm, capable of differentiating into osteoblasts, chondrocytes, adipocytes, and reticular stromal cells. Their critical role in wound healing lies in their secretion of pro-regenerative cytokines and growth factors, such as VEGF and EGF, which help regulate inflammation and stimulate tissue regeneration [14,33].

Recent studies, including those by Hashemi et al. (using amniotic membranes with Wharton jelly-derived MSCs) and Vojtaššák et al. (collagen membranes impregnated with autologous bone marrow MSCs), have demonstrated that MSC administration can accelerate chronic wound healing and reduce wound size, especially in diabetic patients [14,24,33].

However, there are several limitations and challenges, mainly due to difficulties in standardizing production and quality control. Variability in the proliferation and differentiation capacity of MSCs between donors, as well as the presence of distinct subpopulations within the same source, are factors that may influence the efficacy of the therapy. In addition, the origin of MSCs (bone marrow, adipose tissue, umbilical cord) plays a significant role in the results, with specific applications in treating skin wounds or diabetic ulcers [14].

Chen et al. (2022) [29] developed an innovative GelMA/OD/Borax hydrogel with hemostatic and anti-inflammatory properties, which demonstrates efficacy in stopping bleeding and accelerating wound healing due to its tri-network structure and advanced synthesis method. However, the use of Borax and the complexity of UV polymerization raise concerns about potential cytotoxicity, requiring strict production and handling protocols [29,50,52].

Xie et al. (2022) [50] created a hydrogel based on carboxymethyl chitosan (CMCS), sodium alginate (SA), and oxidized dextran (ODE), which rapidly forms a gel through Schiff base and amide reactions. This hydrogel exhibits remarkable hemostatic and antibacterial properties, especially against Staphylococcus aureus. Experiments in animal models (tail amputations and liver injuries in rats) have demonstrated the effective retention of red blood cells in the hydrogel and improved wound healing. However, optimizing the CMCS/ODE/SA ratio and addressing manufacturing costs remain challenges for large-scale implementation [50,52].

Chen et al. (2022) [29] used an in situ crosslinking method to develop a hydrogel composed of CMCS, oxidized dextran, and poly-glutamic acid (γ-PGA). This hydrogel is notable for its ability to absorb excess moisture from the wound surface and electrostatically retain red blood cells, providing effective hemostasis and antibacterial properties. It also promotes tissue regeneration in animal studies. However, the crosslinking parameters and long-term stability require further investigation to prevent unwanted inflammatory reactions and to define optimal dosages [29,52].

Angiogenesis in obese children is improved by diet and lifestyle. El Amrousy et al. (2022) showed that a diet high in omega-3 fatty acids, antioxidants, and other essential nutrients can reduce inflammation and oxidative stress [20], leading to improved endothelial function and the promotion of angiogenesis [9,19,45].

An essential aspect of surgical wound healing is understanding the gender differences in the healing process and scar tissue composition, which are influenced by biological and hormonal factors. Research has highlighted distinct characteristics between male and female skin, both in the composition of the extracellular matrix (ECM) and in the dynamics of the regeneration process.

Male skin demonstrates a greater accumulation of collagen I and ELASTIN in the scar tissue, indicating a denser structure and less flexible ECM regeneration. In contrast, female scars are characterized by an increased accumulation of collagen III, associated with a more regenerative healing process, similar to that observed in fetuses. This trend suggests a greater capacity to maintain skin elasticity and quality after injury [15,31].

In addition, female skin shows higher adipogenic tendencies, confirmed by increased levels of LEPTIN mRNA and other genes related to adipogenesis, indicating better restoration of dermal adipose tissue. In men, these adipogenic tendencies are less pronounced, which may contribute to the increased stiffness and density of their scars [15].

Sex hormones, particularly estrogen, play a significant role in these differences. Estrogen has been associated with better collagen organization and reduced age-related delays in wound healing. Hormone replacement therapy in postmenopausal women has been shown to improve the collagen III/collagen I ratio, thereby improving skin regeneration [31].

Male and female skin respond differently to repair processes following injury. Male scars are characterized by increased density and parallel alignment of collagen I fibers, whereas female scars exhibit a more balanced composition of collagen I and III, suggesting a more flexible and regenerative healing process. These differences present an important opportunity for personalizing wound healing treatments, taking gender into account as a determining factor [15,31].

Frequent physical exercise has a significant effect on vascular health. El Amrousy et al. (2022) revealed that moderate exercise has the potential to improve microcirculation and stimulate the creation of new capillaries in tissues that are affected by obesity [20]. These complementary interventions can support complex therapies with biomaterials or growth factors, helping to heal faster and more effectively [31,54,55]. One of the best ways to improve angiogenesis and wound healing is to reduce weight in obese children through lifestyle and dietary changes. Weight loss not only reduces the adipose tissue layer, but also causes beneficial physiological and metabolic changes that have a positive impact on tissue regeneration processes.

Reducing systemic inflammation is a key benefit of weight loss, particularly in the context of obesity. Pro-inflammatory cytokines, such as TNF-α and IL-6, are released from adipose tissue, contributing to the chronic inflammation observed in obese patients [41,56]. The endothelium is responsible for releasing growth factors, such as VEGF (vascular endothelial growth factor), which are essential for initiating and supporting the process of new capillary formation [29,31,36,42]. Studies show that losing weight reduces oxidative stress, which is another negative factor associated with obesity. This allows the endothelium to function better and support angiogenesis. With decreased inflammation and improved microcirculation, the body is better able to supply affected areas with oxygen and nutrients, which accelerates tissue regeneration [2,12,28,30,37,38,39,40,41,56].

## 5. Challenges and Future Prospectives

The current challenges in wound treatment, particularly in contexts such as humanitarian crises, war zones, and among malnourished or obese children, underscore the urgent need for rapid, effective, and sustainable solutions. Efficient wound healing is critical to reducing complications and mitigating the risk of severe, life-threatening infections. Open or superinfected wounds, common in battlefield settings or due to poor hygiene in humanitarian crises, demand innovative treatment approaches that accelerate regeneration and minimize long-term complications. These situations call for targeted therapies that promote healing while addressing the unique challenges posed by compromised health and environmental factors.

One of the main challenges is the development of treatments that offer both therapeutic efficacy and safety against the backdrop of resource constraints. Nanotherapy and gene therapy are among the most promising approaches in wound healing. Nanomaterials are valued for their ability to reduce inflammation and stimulate regeneration, but high costs and toxicity risks remain major obstacles. Adapting these therapies for use in emergency settings, such as war or humanitarian crises, would require the optimization of dosage and application through controlled delivery methods to ensure long-term efficacy without compromising the safety of vulnerable patients.

In addition, due to its ability to release cytokines and growth factors, which stimulate angiogenesis and tissue regeneration, stem cell therapy offers insights into the healing of complicated wounds. The use of mesenchymal stem cells obtained from multiple sources has been shown to be advantageous in all stages of healing, especially in angiogenesis and re-epithelialization, which are essential processes for persistent or superinfected wounds, such as those occurring in children who do not receive enough nutrition. However, the widespread application of these therapies is limited by cost and issues related to immunogenicity and risk of infection, especially in crisis areas.

Future perspectives focus on the development of therapeutic solutions that combine safety with efficacy by combining advanced nanotherapy technologies and stem cell therapies to promote healing and reduce severe inflammation. Longitudinal studies will be essential to optimize these therapies, given their impact on regeneration in children, immunocompromised patients or in humanitarian crises. The socioeconomic impact of untreated wounds and the potential for these therapies to change current treatments by providing patients with a safe and rapid recovery, even in the most challenging conditions, warrant continued investment in research.

Future studies should include gender as an important variable in research on surgical wound healing in obese patients. Hormonal differences, such as estrogen and testosterone levels, and inflammatory responses significantly influence the healing process.

In addition, the interaction between gender and obesity should be investigated, given the impact of obesity on inflammation and metabolism. Such research could lead to personalized strategies tailored to the specific physiological needs of each gender to improve surgical wound healing.

## 6. Conclusions

In patients with childhood obesity, angiogenic dysfunction presents a critical challenge in the wound healing process, significantly affecting postoperative outcomes and increasing the risk of complications. The reviewed studies highlight chronic low-grade inflammation and oxidative stress as two pivotal factors that impede angiogenesis in these patients. These conditions adversely influence endothelial function and diminish the production of essential growth factors, such as VEGF and PDGF, which are crucial for effective tissue repair. Furthermore, this dysfunction is manifested through disturbances in nitric oxide (NO) signaling, the impaired activity of HIF-2α, and the inadequate role of Tenascin-C in the regeneration of damaged tissues, collectively contributing to delayed and compromised wound healing.

Growth factors, especially VEGF and PDGF, are essential for stimulating and stabilizing new blood vessels. However, in patients with childhood obesity, chronic inflammation and oxidative stress, the activity of these factors is severely impaired. Current treatment methods have been shown to improve angiogenesis and accelerate wound healing, such as the administration of growth factors, the use of multifunctional biomaterials, and gene therapy. However, to avoid the development of unstable or dysfunctional vasculature, special attention is needed in the safety and control of these therapies.

Non-pharmacological interventions such as dietary change and regular exercise have been shown to improve vascular function by reducing inflammation and improving angiogenesis. However, these measures are often insufficient for severe childhood obesity, which requires additional approaches such as the administration of controlled growth factors or the use of gene therapies.

In conclusion, future research must focus on creating safer and more effective treatments that directly address the molecular mechanisms involved in angiogenesis, such as oxidative stress and chronic inflammation. The clinical outcomes of patients with childhood obesity can be significantly improved by implementing these personalized treatments. It is also possible to reduce the risks associated with delayed healing of surgical wounds.

## Figures and Tables

**Figure 1 biomedicines-13-00375-f001:**
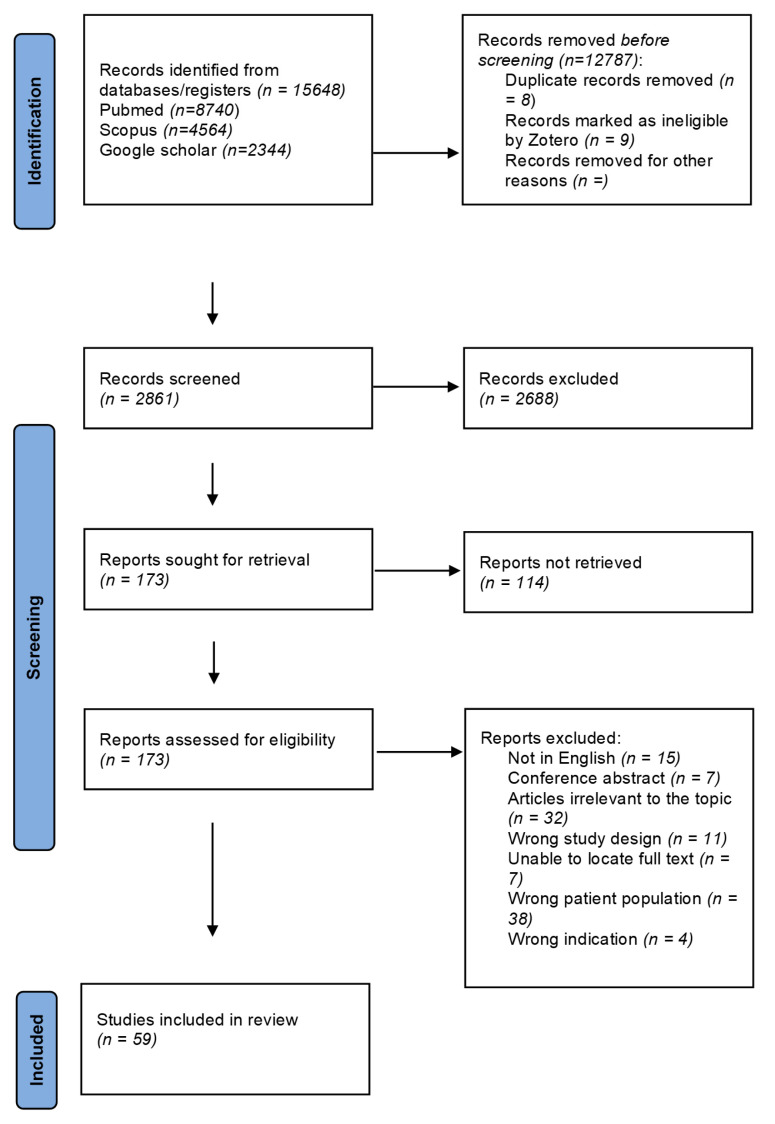
Prisma flow chart; this diagram shows the systematic process we followed to include the works captured by our search.

**Figure 2 biomedicines-13-00375-f002:**
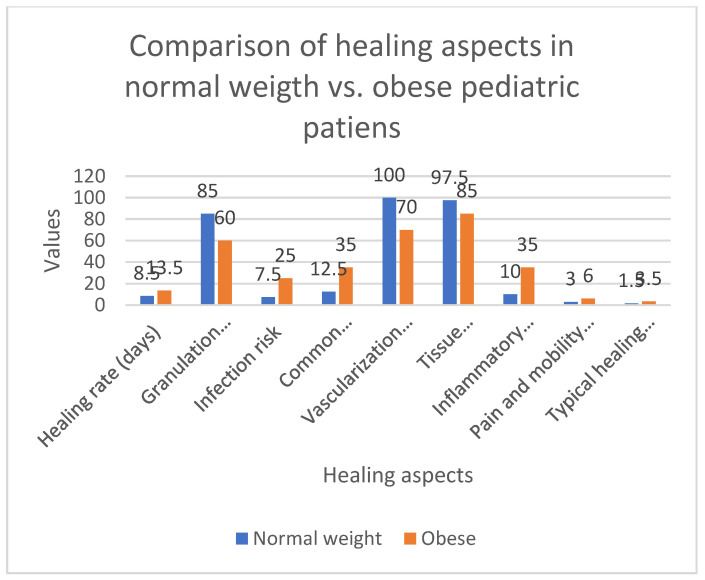
Comparison of healing aspects in normal-weight versus obese pediatric patients. The graph displays several key parameters of wound healing.

**Figure 3 biomedicines-13-00375-f003:**
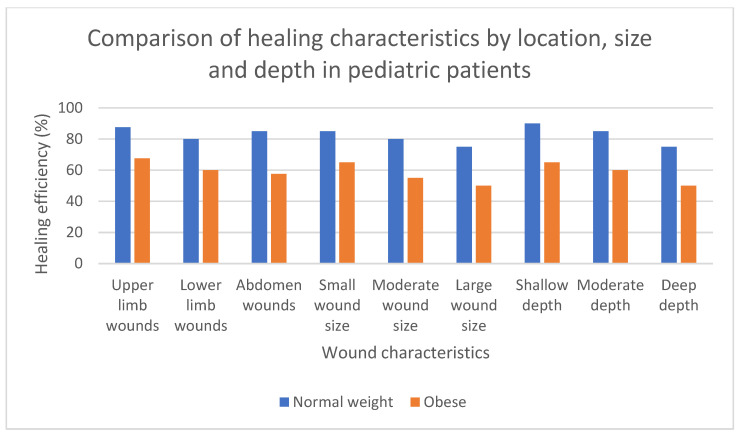
A comparison of healing characteristics by location, size, and depth in pediatric patients. This figure illustrates the healing efficiency (expressed as a percentage) in pediatric patients categorized by normal weight (blue bars) and obesity (orange bars). The analysis considers three wound characteristics: location, size, and depth. Healing locations include wounds on the upper limbs, lower limbs, and abdomen. Wound size is divided into small, moderate, and large wounds, while depth is categorized as shallow, moderate, and deep.

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
