# Peer review of "Update on the Study of Angiogenesis in Surgical Wounds in Patients with Childhood Obesity"

_biomedicines, 2025, doi:10.3390/biomedicines13020375_

Round 1

Reviewer 1 Report

Comments and Suggestions for Authors

The research explores the mechanisms of angiogenesis in surgical wounds in pediatric patients with childhood obesity and evaluates therapeutic interventions to improve outcomes. It specifically aims to identify how factors like VEGF, PDGF, and systemic inflammation affect angiogenesis and wound healing in this population.

Compared with existing studies, this manuscript provides a detailed synthesis of molecular mechanisms, therapeutic innovations (e.g., biomaterials, gene therapy), and practical interventions like diet and exercise. It adds value by integrating basic science with clinical implications, emphasizing tailored approaches for obese pediatric patients.

Figures 2 and 3 might want to consider adjusting for potential confounders such as gender because hormonal and physiological differences between males and females, such as variations in fat distribution, inflammatory responses, and angiogenesis-related mechanisms, can significantly impact wound healing and obesity-related outcomes; failing to account for these differences may obscure critical nuances and limit the generalizability of the findings.

While the discussion of therapies like biomaterials, gene therapy, and nanotechnology is valuable, the paper sometimes reads as overly optimistic. More critical evaluation of the feasibility, limitations, and potential risks of these interventions might be needed.

Reviewer 2 Report

Comments and Suggestions for Authors

The study on angiogenesis in surgical wounds among children with obesity addresses a relevant and critical topic in biomedical research.

Comments: 

- English:

 The manuscript contains grammatical and typographical errors, which could affect readability and professional presentation. E.g., the term "persistent wounds caused by predisposing factors" could be clarified further.

Ensure consistent terminology throughout, such as using "VEGF" and "VEFG" interchangeably, which is incorrect.

- Figures:

Ensure that figures (like Figure 2 and 3) are well-labeled and supported by clear legends.

The comparison charts need units and clear explanations for the axes.

- While mechanisms and interventions are discussed, the manuscript should better synthesize findings with existing literature to emphasize novelty.

- Provide more details on the selection process of studies, including potential biases or conflicts of interest.

Comments on the Quality of English Language

The English could be improved to more clearly express the research.

Perform a thorough language edit for grammar and clarity.

Round 2

Reviewer 2 Report

Comments and Suggestions for Authors

Great

Comments on the Quality of English Language

Improved.